# A Machine Learning Method with Filter-Based Feature Selection for Improved Prediction of Chronic Kidney Disease

**DOI:** 10.3390/bioengineering9080350

**Published:** 2022-07-28

**Authors:** Sarah A. Ebiaredoh-Mienye, Theo G. Swart, Ebenezer Esenogho, Ibomoiye Domor Mienye

**Affiliations:** 1Center for Telecommunications, Department of Electrical and Electronic Engineering Science, University of Johannesburg, Johannesburg 2006, South Africa; snabofa@yahoo.com (S.A.E.-M.); ebenezere@uj.ac.za (E.E.); 2Department of Electrical and Electronic Engineering Science, University of Johannesburg, Johannesburg 2006, South Africa; ibomoiyem@uj.ac.za

**Keywords:** AdaBoost, chronic kidney disease, cost-sensitive learning, machine learning, medical diagnosis

## Abstract

The high prevalence of chronic kidney disease (CKD) is a significant public health concern globally. The condition has a high mortality rate, especially in developing countries. CKD often go undetected since there are no obvious early-stage symptoms. Meanwhile, early detection and on-time clinical intervention are necessary to reduce the disease progression. Machine learning (ML) models can provide an efficient and cost-effective computer-aided diagnosis to assist clinicians in achieving early CKD detection. This research proposed an approach to effectively detect CKD by combining the information-gain-based feature selection technique and a cost-sensitive adaptive boosting (AdaBoost) classifier. An approach like this could save CKD screening time and cost since only a few clinical test attributes would be needed for the diagnosis. The proposed approach was benchmarked against recently proposed CKD prediction methods and well-known classifiers. Among these classifiers, the proposed cost-sensitive AdaBoost trained with the reduced feature set achieved the best classification performance with an accuracy, sensitivity, and specificity of 99.8%, 100%, and 99.8%, respectively. Additionally, the experimental results show that the feature selection positively impacted the performance of the various classifiers. The proposed approach has produced an effective predictive model for CKD diagnosis and could be applied to more imbalanced medical datasets for effective disease detection.

## 1. Introduction

Chronic kidney disease is among the leading causes of death globally. A recent medical report states that approximately 324 million people suffer from CKD globally [1]. The glomerular filtration rate (GFR) is a widely used CKD screening test [2]. Though CKD affects people worldwide, it is more prevalent in developing countries [3]. Meanwhile, early detection is vital in reducing the progression of CKD. However, people from developing countries have not benefitted from early-stage CKD screening due to the cost of diagnosing the disease and limited healthcare infrastructure. While the global prevalence of CKD is reported to be 13.4% [4], it is said to have a 13.9% prevalence in Sub-Saharan Africa [5,6]. Another study reported a 16% pooled prevalence of CKD in West Africa [7], the highest in Africa. Numerous research works have specified that CKD is more prevalent in developing countries [8]. Notably, it is reported that 1 out of every 10 persons suffers from CKD in South Asia, including Pakistan, India, Bhutan, Bangladesh, and Nepal [3].

Therefore, several researchers have proposed machine learning (ML)-based methods for the early detection of CKD. These ML methods could provide effective, convenient, and low-cost computer-aided CKD diagnosis systems to enable early detection and intervention, especially in developing countries. Researchers have proposed different methods to detect CKD effectively using the CKD dataset [9] available at the University of California, Irvine (UCI) machine learning repository. For example, Qin et al. [9] proposed an ML approach for the early detection of CKD. The approach involved using the k-Nearest Neighbor (KNN) imputation to handle the missing values in the dataset. After filling the missing data, six ML classifiers were trained and tested with the preprocessed data. The classifiers include logistic regression, SVM, random forest, KNN, naïve Bayes, and a feed-forward neural network. Due to the misclassification of these classifiers, the authors developed an integrated classifier that uses a perceptron to combine the random forest and logistic regression classifiers, which produced an enhanced accuracy of 99.83%.

Meanwhile, Ebiaredoh-Mienye et al. [10] proposed a method for improved medical diagnosis using an improved sparse autoencoder (SAE) network with a softmax layer. The neural network achieved sparsity through weight penalty, unlike the traditional sparse autoencoders that penalize the hidden layer activations. When used for CKD prediction, the proposed SAE achieved an accuracy of 98%. Chittora et al. [11] studied how to effectively diagnose CKD using ML methods. The research employed seven algorithms, including the C5.0 decision tree, logistic regression, linear support vector machine (LSVM) with 𝐿1 and 𝐿2 norms, artificial neural network, etc. The authors studied the performance of the selected classifiers when they were trained under different experimental conditions. These conditions include instances where the complete and reduced feature sets were used to train the classifiers. The experimental results showed that the LSVM with L2 norm trained with the reduced feature set obtained an accuracy of 98.46%, which outperformed the other classifiers.

Furthermore, Silveira et al. [12] developed a CKD prediction approach using a variety of resampling techniques and ML algorithms. The resampling techniques include the synthetic minority oversampling technique (SMOTE) and Borderline-SMOTE, while the classifiers include random forest, decision tree, and AdaBoost. The experimental results showed that the decision tree with the SMOTE technique achieved the best performance with 98.99%. Generally, these ML research works utilize many attributes such as albumin, hemoglobin level, white blood cell count, red blood cell count, packed cell volume, blood pressure, specific gravity, etc., to flag patients at risk of CKD, thereby allowing clinicians to provide early and cost-efficient medical intervention. Despite the attention given to CKD prediction using machine learning, only a few research works have focused on identifying the most relevant features needed to improve CKD detection [13,14,15]. If identified correctly in suspected CKD patients, these features could be utilized for efficient computer-aided CKD diagnosis.

In machine learning tasks, the algorithms employ discriminative abilities of features in classifying the samples. The ML models’ performance relies not only on the specific training algorithm but also on the input data characteristics, such as the number of features and the correlation between the features [16]. Moreover, in most ML applications, especially in medical diagnosis, all the input features may not have equal importance. The goal of feature selection is to remove redundant attributes from the input data, ensuring the training algorithm learns the data more effectively. By removing non-informative variables, the computational cost of building the model is reduced, leading to faster and more efficient learning with enhanced classification performance.

Filter- and wrapper-based methods are the two widely used feature selection mechanisms [17]. Wrapper-based feature selection techniques use a classifier to build ML models with different predictor variables and select the variable subset that leads to the best model. In contrast, filter-based methods are statistical techniques independent of a learning algorithm used to compute the correlation between the predictor and independent variables [18]. The predictor variables are scored according to their relevance to the target variable. The variables with higher scores are then used to build the ML model. Therefore, this research aims to use information gain (IG), a filter-based feature selection method, to identify the most relevant features for improved CKD detection. IG is a robust algorithm for evaluating the gain of the various features with respect to the target variable [19]. The attributes with the least IG values are removed, and those whose IG values are above a particular threshold are used to train the classifiers.

Meanwhile, a significant challenge in applying machine learning algorithms for medical diagnosis is the imbalanced class problem [20,21]. Most ML classifiers underperform when trained with imbalanced datasets. Class imbalance implies there is an uneven distribution of samples in each class. The class with the most samples is the majority class, while the class with the lesser samples is the minority class. Imbalance learning can be divided into data and algorithm-level approaches [22]. Data level methods are based on resampling techniques. Several studies have employed resampling techniques such as undersampling and oversampling to solve the class imbalance problem [23,24,25]. In order to create a balanced dataset, undersampling methods remove samples from the majority class, while oversampling techniques artificially create and add more data in the minority class. However, there are limitations to using these resampling techniques. For example, the samples discarded from the majority class could be vital in efficiently training the classifiers [20]. Therefore, several studies have resorted to using algorithm-level methods such as ensemble learning and cost-sensitive learning to effectively handle the imbalanced data instead of data-level techniques [26,27,28,29].

Ensemble learning is a breakthrough in ML research and application, which is used to obtain a very accurate classifier by combining two or more classifiers. Boosting [30] and Bagging [31] are widely used ensemble learning techniques. Adaptive boosting [30] is a type of boosting technique that creates many classifiers by assigning weights to the training data and adjusting these weights after every boosting cycle. The wrongly classified training instances are given higher weights in the next iteration, whereas the weight of correctly predicted examples is decreased. However, the AdaBoost algorithm does not treat the minority class and majority class weight updates differently when faced with imbalanced data. Therefore, in this study, we develop an AdaBoost classifier that gives higher weight to examples in the minority class, thereby enhancing the prediction of the minority class samples and the overall classification performance. A cost-sensitive classifier is obtained by biasing the weighting technique to focus more on the minority class. Recent findings have demonstrated that cost-sensitive learning is an efficient technique suitable for imbalanced classification problems [32,33,34]. The contribution of this study is to obtain the most important CKD attributes needed to improve the performance of CKD detection and develop a cost-sensitive AdaBoost classifier that gives more attention to samples in the minority class.

The rest of this paper is structured as follows: Section 2 presents the materials and methods, including an overview of the CKD dataset, the information gain technique, the traditional AdaBoost method, the proposed cost-sensitive AdaBoost, and the performance evaluation metrics. Section 3 presents the experimental results and discussion, while Section 4 concludes the paper.

## 2. Materials and Methods

This section provides an overview of the CKD dataset and the various methods used in the research. In particular, a detailed overview of the traditional AdaBoost algorithm and the proposed cost-sensitive AdaBoost is presented, thereby showing the difference between both methods.

### 2.1. Dataset

This study utilizes the CKD dataset prepared in 2015 by Apollo Hospitals, Tamil Nadu, India. The dataset is publicly available at the University of California, Irvine (UCI) machine learning repository [35]. It contains medical test results and records from 400 patients; 250 correspond to patients with CKD, and 150 correspond to patients without CKD, so the dataset is imbalanced. There are 24 independent variables (11 numerical and 13 nominal) and a class variable (ckd or notckd). The attributes and their corresponding descriptions are shown in Table 1.

Some of the features in Table 1 are briefly described as follows: Specific gravity estimates the concentration of particles in the urine and the urine’s density relative to the density of water. It indicates the hydration status of a patient together with the functional ability of the patient’s kidney. Albumin is a protein found in the blood [36]. When the kidney is damaged, it allows albumin into the urine. Higher albumin levels in the urine could indicate the presence of CKD. Meanwhile, blood urea indicates vital information about the functionality of the kidney. A blood urea nitrogen test measures the quantity of urea nitrogen in a patient’s blood, and a high amount implies the kidneys are not functioning normally. While a random blood glucose test measures the amount of sugar circulating in a patient’s blood, and a level of 200 mg/dL or above implies the patient has diabetes. Serum creatinine is a waste product produced by a person’s muscles [37]. A creatinine test measures the creatinine levels in the blood or urine, and high levels of the substance imply the kidney is not functioning well enough to filter the waste from the blood.

Furthermore, sodium is an electrolyte in the blood that helps the muscles and nerves work effectively. A sodium blood test measures the amount of sodium in the patient’s blood, and a very high or low amount may indicate a kidney problem, dehydration, or other medical condition. Potassium is another electrolyte in the blood, and a very high or low amount could signal the presence of an underlying condition. White blood cells (WBC) protect the human body from invading pathogens. They are part of the body’s immune system, protecting it from infections [38]. The normal range is between 4000 and 11,000 per microliter of blood. Elevated WBC count is a popular indicator of the progression of CKD. Red blood cells (RBC) in humans deliver oxygen to the body tissues. The average RBC count is 4.7 to 6.1 million cells per microliter for men and 4.2 to 5.4 million cells per microliter for women. A low RBC, also called anemia, is a common complication of CKD.

Meanwhile, the data needs to be preprocessed to make it suitable for machine learning. Therefore, all the nominal or categorical data were coded. Specifically, the attributes whose scales are ‘normal’ and ‘abnormal’ were transformed to 1 and 0, respectively. The attributes whose scales are ‘present’ and ‘not present’ were transformed to 1 and 0, respectively. Additionally, the ‘yes’ and ‘no’ scales were coded to 1 and 0, respectively. Lastly, the attribute with ‘good’ and ‘poor’ scales was transformed to 1 and 0, respectively.

Furthermore, the dataset contains a few missing values. It is vital to appropriately deal with missing values before building ML models because ignoring or deleting the missing data could degrade or bias the performance of the models [39,40]. Imputation is a method used to estimate and fill missing values in a dataset. Since the number of missing values in our dataset is not large, the mean imputation technique is used to handle the missing values. The mean imputation technique computes the mean of the observed values for each variable, and the missing values are filled with the corresponding computed mean value [41]. Meanwhile, except for the ‘age’ and binary attributes, the remaining attributes were scaled to have values between 0 and 1 using the Min–Max Scaling technique [42].

Additionally, the clinicians at Apollo Hospitals, Tamil Nadu, India, categorized attributes as normal or abnormal, present or not present, yes or no. The clinicians selected the 24 attributes representing the patient’s medical tests and records associated with chronic kidney disease. However, the attributes do not carry equal weights in diagnosing a patient with CKD, and some attributes are more indicative of the presence of CKD than others. Additionally, certain attributes might be redundant and could increase the complexity of the ML model [43]. Hence, this research employs the information gain technique to rank the attributes according to their relevance in detecting the disease, and only the most relevant attributes are used to build the ML model.

### 2.2. Information Gain

Effective feature selection could remove attributes that are less useful in obtaining an excellent predictive model. Additionally, it is necessary to remove attributes unrelated to the target variable because these attributes could increase the computational cost and prevent the model from obtaining optimal performance [44]. This study utilizes the information gain (IG) technique to extract the optimal features. IG is a type of filter-based feature selection that calculates the predictor variable’s ability to classify the dependent variable [45]. The IG method has its roots in information theory, and it calculates the statistical dependence between two variables. Mathematically, the IG between two variables X and Y is formulated as:(1)$$\mathrm{IG}\left(X|Y\right)=H\left(X\right)-H\left(X|Y\right),$$
where $H\left(X\right)$ is the entropy for variable X and $H\left(X|Y\right)$ represents the conditional entropy for X given Y. Computing the IG value for an attribute involves calculating the entropy of the target variable for the whole dataset and subtracting the conditional entropies for every potential value of that attribute [46]. Furthermore, the entropy $H\left(X\right)$ and conditional entropy $H\left(X|Y\right)$ are computed as:(2)$$H\left(X\right)=-\sum_{x\in X}P\left(x\right)log_{2}\left(x\right),$$
(3)$$H\left(X|Y\right)=-\underset{x\in X}{\sum }P\left(x\right)\underset{y\in Y}{\sum }P\left(x|y\right)log_{2}\left(P\left(x|y\right)\right),$$

Hence, given two variables X and Z, a given variable Y is said to have a more significant correlation to X than Z if $\mathrm{IG}\left(X|Y\right)>\mathrm{IG}\left(Z|Y\right)$. Furthermore, IG considers every attribute in isolation, calculates its information gain, and computes its relevance to the target variable.

### 2.3. AdaBoost Algorithm

The AdaBoost algorithm is an ML technique derived from the concept of boosting. The boosting technique primarily entails converting weak learners into strong learners [47]. Freund and Schapire [48] proposed the AdaBoost algorithm to iteratively train multiple learning classifiers using the same training dataset. After training the weak learners, they are combined to obtain a strong classifier. The AdaBoost procedure involves selecting an appropriate weak learner and employing the same training dataset to train the weak learner iteratively to enhance its performance, as shown in Algorithm 1. Two weights are utilized in implementing the AdaBoost algorithm; the first is the sample weight, and the second is the weight of every weak learner [49]. The algorithm adjusts the sample weight depending on the weak classifier’s result, thereby giving more attention to wrongly classified samples. Subsequent base learners are trained with the adjusted samples [50]. The final strong classifier is obtained by combining the output of the weak learners using a weighted sum [51]. The AdaBoost is adaptive because subsequent weak classifiers are trained to pay more attention to the samples that were wrongly classified by preceding classifiers.
**Algorithm 1:** Conventional AdaBoost technique**Input:** training dataset $S=\left\{\left(x_{1},y_{1}\right),\ldots ,\left(x_{2},y_{2}\right),\ldots ,\left(x_{n},y_{n}\right)\right\}$, base learner h, the number of training rounds T.**Output:** the final strong classifier H.**Procedure:**1.***for*** i=1:1:n2.compute the weight of the sample $x_{i}$: $D_{1}\left(i\right)=\frac{1}{n}$3.***end for***4.***for*** t=1:1:T5.select a training data subset X from S, fit h using X to get a weak classifier $h_{t}$, compute the classification error $\varepsilon_{t}$: $\varepsilon_{t}=P\left[h_{t}\left(x_{i}\right)\neq y_{i}\right]=\overset{n}{\underset{i=1}{\sum }}D_{t}\left(i\right)Ι\left[h_{t}\left(x_{i}\right)\neq y_{i}\right]$where $h_{t}\left(x_{i}\right)$ denotes the predicted label of $x_{i}$ using the weak classifier $h_{t}$, and $y_{i}$ denotes the actual label of $x_{i}$.6.compute the weight of $h_{t}:\;\alpha_{t}=\frac{1}{2}\ln\left(\frac{1-\varepsilon_{t}}{\varepsilon_{t}}\right)$7.update the weight of all the instances in S:***for*** i=1:1:n$D_{t+1}\left(i\right)=\frac{D_{t}\left(i\right)}{Z_{t}}\exp\left(-\alpha_{t}y_{i}h_{t}\left(x_{i}\right)\right)$where $Z_{t}$ is a normalization factor and is calculated as: $Z_{t}=\overset{n}{\underset{i=1}{\sum }}D_{t}\left(i\right)\exp\left(-\alpha_{t}y_{i}h_{t}\left(x_{i}\right)\right)$
8.***end for***9.***end for***10.assuming $H\left(x\right)$ is the class label for an instance x; after the iterations, the final classifier H is obtained as: $H\left(x\right)=sign\left(\overset{T}{\underset{t=1}{\sum }}\alpha_{t}h_{t}\left(x\right)\right)$

### 2.4. Proposed Cost-Sensitive AdaBoost

At every iteration, the AdaBoost algorithm increases the weights of the misclassified training samples and decreases that of the samples that were predicted correctly. This weighting method distinguishes instances as correctly or wrongly classified, and the examples from both classes are treated equally. For example, the weights of incorrectly classified instances from both classes are increased by a similar ratio. The weights of the samples that were predicted correctly from both classes are decreased by a similar ratio [52]. However, in imbalance classification, the goal is to improve the classifier’s prediction performance on the minority class. Hence, a suitable weighting approach will identify the different types of instances and give more weight to the samples with greater detection importance, i.e., the minority class.

From the AdaBoost learning method (Algorithm 1), cost items ($\beta_{i}$) are added to the weight update equation to bias the weighting technique. The rate of correct and incorrect predictions of the various classes are included as part of $\beta_{i}$. The cost of false positive is denoted as $c_{10}$ and the cost of false negative is denoted as $c_{01}$, while the cost of true positive and true negative are denoted as $c_{11}$ and $c_{00}$, respectively [53]. The weight update in the traditional AdaBoost takes into account the overall error rate only [54]. However, in the cost-sensitive AdaBoost, the error rate of each class is considered. In this new weighting strategy, the weights of the minority class examples are higher compared to the majority class examples. The new cost-sensitive AdaBoost is presented in Algorithm 2.
**Algorithm 2:** Cost-Sensitive AdaBoost**Input:** training dataset $S=\left\{\left(x_{1},y_{1}\right),\ldots ,\left(x_{2},y_{2}\right),\ldots ,\left(x_{n},y_{n}\right)\right\}$, base learner h, the number of iterations T.**Output:** the final strong classifier H.**Procedure:**1.***for*** i=1:1:n2.compute the weight of the sample $x_{i}$: $D_{1}\left(i\right)=\frac{1}{n}$3.***end for***4.***for*** t=1:1:T5.select a training data subset X from S, fit h using X to get a weak classifier $h_{t}$6.let n+ and n− indicate the positive and negative classes, respectively. Compute the error rate $\varepsilon_{t}$ of the base learner for both the positive class $\varepsilon_{t}^{p}$ and negative class $\varepsilon_{t}^{n}$: $\varepsilon_{t}=\left(\frac{\varepsilon_{t}^{p}+\varepsilon_{t}^{n}}{2}\right),\;$ where   $\varepsilon_{t}^{p}=P\left[h_{t}\left(x_{i}\right)\neq y_{i}\right]=\overset{n+}{\underset{i=1}{\sum }}D_{t}\left(i\right)Ι\left[h_{t}\left(x_{i}\right)\neq y_{i}\right]\;\mathrm{and}\;\varepsilon_{t}^{n}=P\left[h_{t}\left(x_{i}\right)\neq y_{i}\right]=\overset{n-}{\underset{i=1}{\sum }}D_{t}\left(i\right)Ι\left[h_{t}\left(x_{i}\right)\neq y_{i}\right]$7.compute the weight of $h_{t}$: $\alpha_{t}=\frac{1}{2}\ln\left(\frac{1-\varepsilon_{t}}{\varepsilon_{t}}\right)$8.update the weight of all the instances in S:***for*** i=1:1:n          $D_{t+1}\left(i\right)=\frac{D_{t}\left(i\right)}{Z_{t}}\exp\left(-\alpha_{t}\beta_{i}y_{i}h_{t}\left(x_{i}\right)\right)$where $Z_{t}$ is a normalization factor and is calculated as: $Z_{t}=\overset{n}{\underset{i=1}{\sum }}D_{t}\left(i\right)\exp\left(-\alpha_{t}\beta_{i}y_{i}h_{t}\left(x_{i}\right)\right)$and $\beta_{i}$ is calculated as: $\beta_{i}=\left\{\begin{array}{c} \frac{TP_{t}}{FP_{t}+TP_{t}}c_{10},\;if\;y_{i}=1,\;h_{t}\left(x_{i}\right)=-1\\ \frac{TN_{t}}{FN_{t}+TN_{t}}c_{01},\;if\;y_{i}=-1,\;h_{t}\left(x_{i}\right)=1\\ \frac{TP_{t}}{FP_{t}+TP_{t}}c_{11},\;if\;y_{i}=1,\;h_{t}\left(x_{i}\right)=1\\ \frac{TN_{t}}{FN_{t}+TN_{t}}c_{00},\;if\;y_{i}=-1,\;h_{t}\left(x_{i}\right)=-1 \end{array}\right.$where $TP_{t}$, $TN_{t}$, $FP_{t}$, $FN_{t}$ are true positive, true negative, false positive, and false negative values for iteration t. Meanwhile, $c_{10}$, $c_{01}$, $c_{11}$, $c_{00}$ are the cost-sensitive factors, where $c_{10}>c_{00}$ and $c_{01}>c_{11}$.9.***end for***10.***end for***11.the final classifier is obtained as follows: $H\left(x\right)=sign\left(\overset{T}{\underset{t=1}{\sum }}\alpha_{t}h_{t}\left(x\right)\right)$

By giving higher weights to samples in the minority class, the weak classifiers tend to focus more on the misclassification of examples in that class, thereby accurately classifying more instances at each iteration. Therefore, the final strong classifier will obtain more correct predictions. The architecture of the proposed approach is shown in Figure 1.

### 2.5. Performance Evaluation Metrics

The dataset used in this research comprises two classes, ckd and notckd classes. The ckd-labeled data are the positive patients, while the notckd-labeled data are the negative patients. Meanwhile, accuracy (ACC), sensitivity (SEN), and specificity (SPE) are used to assess the performance of the classifiers. Accuracy is the total number of correct predictions divided by the total number of predictions made by the classifier. Sensitivity or true positive rate is the number of correct positive predictions divided by the number of positive cases in the dataset; it measures the ability of the classifier to correctly detect those with the disease [55]. Specificity measures the classifier’s ability to correctly identify people without the disease, i.e., negative instances. These metrics are computed as follows: (4)$$\mathrm{Accuracy}=\frac{TP+TN}{TP+TN+FP+FN}\;,$$
(5)$$\mathrm{Sensitivity}=\frac{TP}{TP+FN}\;,$$
(6)$$\mathrm{Specificity}=\frac{TN}{TN+FP}\;,$$
where true positive (TP) represents the ckd instances that were correctly classified, and false-negative (FN) represents the ckd instances that were wrongly classified. True negative (TN) denotes the notckd samples that were correctly classified, and false-positive (FP) indicates the notckd instances that were wrongly classified [56]. Additionally, this research utilizes the receiver operating characteristic (ROC) curve and the area under the ROC curve (AUC) to further evaluate the classifiers’ performance. The ROC curve is a plot of the true positive rate (TPR) against the false positive rate (FPR) at different threshold values. It shows the ability of the classifier to distinguish between the ckd and notckd classes. Meanwhile, the AUC is mainly utilized to summarize the ROC curve, and it has a value between 0 and 1 that shows the classifiers’ ability to differentiate between both classes [57]. The higher the AUC value, the better the classifiers can distinguish between the ckd and notckd classes.

## 3. Results

In this section, the experimental results are presented and discussed. All experiments used the preprocessed data, as discussed in Section 2.1, and the ML models were developed using scikit-learn [58], a machine learning library for Python programming. The experiments were conducted using a computer with the following specifications: Intel(R) Core(TM) i7-113H @ 3.30 GHz, 4 Core(s), and 16 GB RAM. Furthermore, to have a baseline for comparing the proposed cost-sensitive AdaBoost (CS AdaBoost), this research implements the traditional AdaBoost [30] presented in Algorithm 1 and other well-known classifiers, including logistic regression [59], decision tree [60], XGBoost [61], random forest [62], and SVM [63]. The classifiers are trained with the complete feature set and the reduced feature set to demonstrate the impact of the feature selection. Meanwhile, the 10-fold cross-validation method is employed to evaluate the performance of the various models. The decision tree algorithm is the base learner for the AdaBoost and CS AdaBoost implementations.

### 3.1. Performance of the Classifiers without Feature Selection

This subsection presents the experimental results obtained when the complete feature set was used to train the various classifiers. These results are tabulated in Table 2. Additionally, Figure 2 shows the AUC values and the ROC curves of the different classifiers.

Table 2 and Figure 2 show that the proposed cost-sensitive AdaBoost obtained excellent performance by outperforming the traditional AdaBoost and the other classifiers, having obtained an AUC, accuracy, sensitivity, and specificity of 0.980, 0.967, 0.975, and 0.960, respectively.

### 3.2. Performance of the Classifiers after Feature Selection

The information-gain-based feature selection ranked the chronic kidney disease attributes. This step aims to select the features with the highest information gain with respect to the target variable. The ranked features and their IG values are shown in Table 3. After obtaining the IG values of the various features, the standard deviation [19] of the values is computed, which serves as the threshold value for the feature selection. The standard deviation measure has been used in recent research to obtain a reasonable threshold for feature selection [19,64,65]. The threshold value obtained is 0.156. Therefore, the IG values equal to or greater than 0.156 are selected as the informative features and used for building the models. In contrast, the attributes with IG values lower than the threshold are discarded. Hence, from Table 3, the top 18 features are selected as the optimal feature set since their IG values (rounded to three decimal values) are greater than 0.156, and the following features are discarded: f22, f5, f24, f8, f9, and f21.

To demonstrate the effectiveness of the feature selection, the reduced feature set is used to train the proposed CS AdaBoost and the other classifiers. The experimental results are shown in Table 4. Additionally, the ROC curve and the various AUC values are shown in Figure 3. The experimental results in Table 4 and Figure 3 show that the proposed CS AdaBoost obtained an AUC, accuracy, sensitivity, and specificity of 0.990, 0.998, 1.000, and 0.998, respectively, which outperformed the logistic regression, decision tree, XGBoost, random forest, SVM, and conventional AdaBoost. Secondly, it is observed that the performance of the various classifiers in Table 4 is better than their corresponding performance in Table 2. This improvement demonstrates the effectiveness of the feature selection step. Therefore, the combination of feature selection and cost-sensitive AdaBoost is an effective method for predicting CKD.

### 3.3. Comparison with Other CKD Prediction Studies

Even though the proposed approach showed superior performance to the other algorithms, it is not enough to conclude its robustness. It is, however, necessary to compare it with other state-of-the-art methods in the literature. Hence, the proposed approach is compared with the following methods: a probabilistic neural network (PNN) [66], an enhanced sparse autoencoder (SAE) neural network [10], a naïve Bayes (NB) classifier with feature selection [67], a feature selection method based on cost-sensitive ensemble and random forest [3], a linear support vector machine (LSVM) and synthetic minority oversampling technique (SMOTE) [11], a cost-sensitive random forest [68], a feature selection method based on recursive feature elimination (RFE) and artificial neural network (ANN) [69], a correlation-based feature selection (CFS) and ANN [69]. The other methods include optimal subset regression (OSR) and random forest [9], an approach to identify the essential CKD features using improved linear discriminant analysis (LDA) [13], a deep belief network (DBN) with Softmax classifier [70], a random forest (RF) classifier with feature selection (FS) [71], a model based on decision tree and the SMOTE technique [12], a logistic regression (LR) classifier with recursive feature elimination (RFE) technique [14], and an XGBoost model with a feature selection approach combining the extra tree classifier (ETC), univariate selection (US), and RFE [15].

The proposed approach based on cost-sensitive AdaBoost and feature selection achieved excellent performance compared to several state-of-the-art methods in the literature, as shown in Table 5.

### 3.4. Discussion

This study aimed to solve two problems: first, to select the most informative features to enable the effective detection of CKD. The second aim was to develop an effective cost-sensitive AdaBoost classifier that accurately classifies samples in the minority class. The use of more features than necessary sometimes affects ML classifiers’ performance and increases the computational cost of training the models. Hence, this research employed the IG-based feature selection method to obtain the optimal feature set. Furthermore, seven classifiers were used in this study, trained using the complete and the reduced feature sets. From the experimental results, the proposed framework showed improved classification performance with the reduced feature set, i.e., 18 out of 24 input variables. Additionally, the models trained with the reduced feature set performed better than those trained with the complete feature set. Remarkably, the proposed method obtained higher performance than the other classifiers.

Furthermore, the features selected by the IG technique were similar to current medical practices. For example, the IG technique ranked albumin, hemoglobin, packed cell volume, red blood cell count, and serum creatinine as the most informative features, and numerous studies have identified a strong correlation between these variables and chronic kidney disease [71,72,73,74].

Meanwhile, the class imbalance problem is common in most real-world classification tasks. Another objective of this study was to develop a robust classifier to prevent the misclassification of the minority class that occurs when classifiers are trained using imbalanced data. Hence, this study developed a cost-sensitive AdaBoost classifier, giving more attention to the minority class. The experimental results indicate that the proposed method achieved a higher classification performance than the baseline classifiers and techniques in recent literature. Secondly, the results demonstrate that the combination of the information-gain-based feature selection and the cost-sensitive AdaBoost classifier significantly improved the detection of chronic kidney disease.

## 4. Conclusions

This paper proposed an approach that combines information-gain-based feature selection and a cost-sensitive AdaBoost classifier to improve the detection of chronic kidney disease. Six other machine learning classifiers were implemented as the baseline for performance comparison. The classifiers include logistic regression, decision tree, random forest, SVM, XGBoost, and the traditional AdaBoost. Firstly, the IG technique was used to compute and rank the importance of the various attributes. Secondly, the classifiers were trained with the reduced and complete feature sets. The experimental results show that selected features enhanced the performance of the classifiers.

Furthermore, the proposed cost-sensitive AdaBoost achieved superior performance to the other classifiers and methods in recent literature. Therefore, combining the IG-based feature selection technique and cost-sensitive AdaBoost is an effective approach for CKD detection and can be potentially applied for early detection of CKD through computer-aided diagnosis. Future research will focus on collecting large amounts of data to train ML models, including datasets that allow for the prediction of the disease severity, duration of the disease, and the age of onset.

## Figures and Tables

**Figure 1 bioengineering-09-00350-f001:**
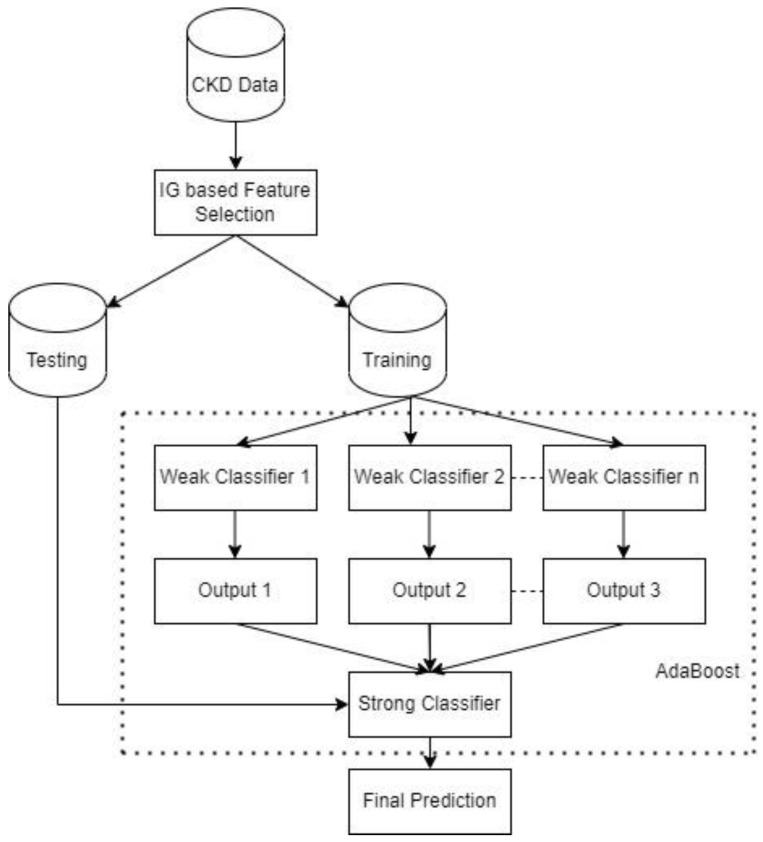
The architecture of the proposed approach.

**Figure 2 bioengineering-09-00350-f002:**
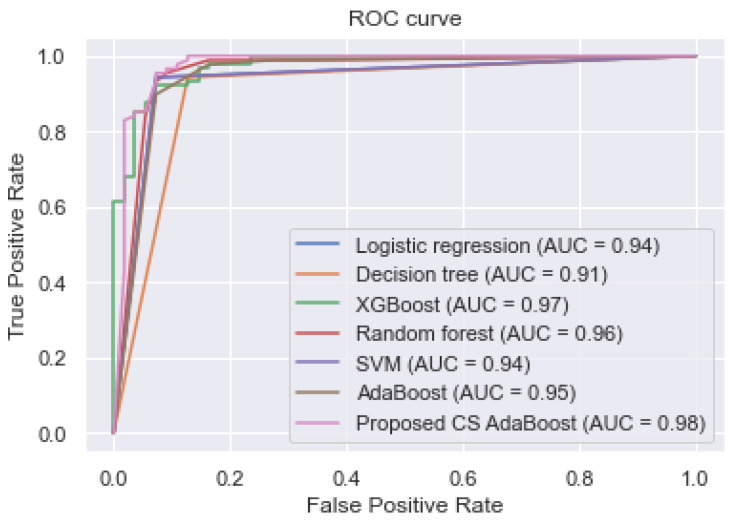
ROC curve of the classifiers trained using the complete feature set.

**Figure 3 bioengineering-09-00350-f003:**
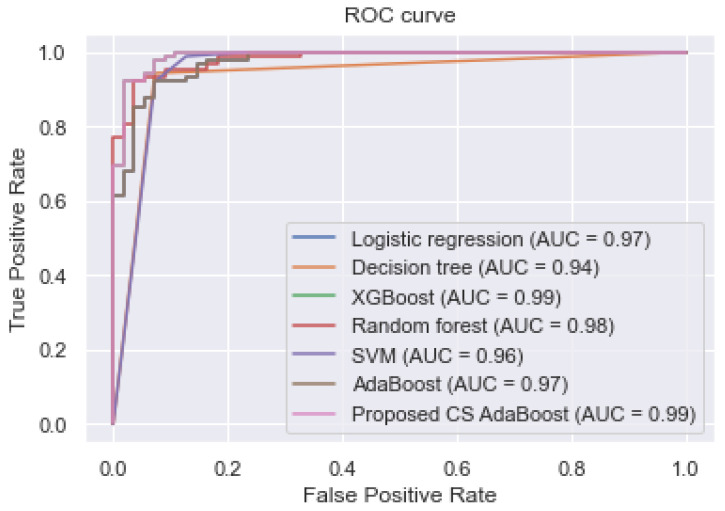
ROC curve of the classifiers trained with the reduced feature set.

**Table 1 bioengineering-09-00350-t001:** CKD dataset description.

No.	Attribute	Description	Category	Scale
f1	age	Age of the patient	Numerical	age in years
f2	bp	Blood pressure	Numerical	mm/Hg
f3	sg	Specific gravity	Nominal	1.005, 1.010, 1.015, 1.020, 1.025
f4	al	Albumin	Nominal	0, 1, 2, 3, 4, 5
f5	su	Sugar	Nominal	0, 1, 2, 3, 4, 5
f6	rbc	Red blood cells	Nominal	normal, abnormal
f7	pc	Pus cell	Nominal	normal, abnormal
f8	pcc	Pus cell clumps	Nominal	present, not present
f9	ba	Bacteria	Nominal	present, not present
f10	bgr	Blood glucose random	Numerical	mgs/dl
f11	bu	Blood urea	Numerical	mgs/dl
f12	sc	Serum creatinine	Numerical	mgs/dl
f13	sod	Sodium	Numerical	mEq/L
f14	pot	Potassium	Numerical	mEq/L
f15	hemo	Hemoglobin	Numerical	gms
f16	pcv	Packed cell volume	Numerical	-
f17	wc	White blood cell count	Numerical	cells/cumm
f18	rc	Red blood cell count	Numerical	millions/cmm
f19	htn	Hypertension	Nominal	yes, no
f20	dm	Diabetes mellitus	Nominal	yes, no
f21	cad	Coronary artery disease	Nominal	yes, no
f22	appet	Appetite	Nominal	good, poor
f23	pe	Pedal edema	Nominal	yes, no
f24	ane	Anemia	Nominal	yes, no
f25	class	Class	Nominal	ckd, notckd

**Table 2 bioengineering-09-00350-t002:** Performance of the classifiers trained with the complete feature set.

Classifier	ACC	SEN	SPE	AUC
Logistic regression	0.940	0.948	0.933	0.940
Decision tree	0.902	0.932	0.890	0.910
XGBoost	0.958	0.964	0.942	0.970
Random forest	0.952	0.955	0.940	0.960
SVM	0.937	0.943	0.930	0.940
AdaBoost	0.930	0.941	0.935	0.950
Proposed CS AdaBoost	0.967	0.975	0.960	0.980

**Table 3 bioengineering-09-00350-t003:** Feature ranking.

No.	Feature Name	IG Value
f4	al	0.598
f15	hemo	0.581
f16	pcv	0.526
f18	rc	0.482
f12	sc	0.474
f10	bgr	0.422
f3	sg	0.392
f11	bu	0.389
f13	sod	0.344
f17	wc	0.325
f19	htn	0.270
f14	pot	0.266
f1	age	0.253
f7	pc	0.251
f20	dm	0.215
f2	bp	0.209
f6	rbc	0.206
f23	pe	0.184
f22	appet	0.155
f5	su	0.135
f24	ane	0.128
f8	pcc	0.097
f9	ba	0.069
f21	cad	0.065

**Table 4 bioengineering-09-00350-t004:** Performance of the classifiers trained with the reduced feature set.

Classifier	ACC	SEN	SPE	AUC
Logistic regression	0.961	0.959	0.961	0.970
Decision tree	0.940	0.935	0.948	0.940
XGBoost	0.989	0.990	0.986	0.990
Random forest	0.977	0.981	0.973	0.980
SVM	0.954	0.957	0.961	0.960
AdaBoost	0.964	0.960	0.968	0.970
Proposed CS AdaBoost	0.998	1.000	0.998	0.990

**Table 5 bioengineering-09-00350-t005:** Comparison with other studies.

Reference	Method	ACC	SEN	SPE	AUC
Rady and Anwar [66]	PNN	0.969	0.987	0.964	-
Ebiaredoh-Mienye et al. [10]	SAE	0.980	0.970	-	-
Almustafa [67]	NB and FS	0.976	0.988	-	0.989
Ali et al. [3]	Cost-sensitive ensemble with RF	0.967	0.986	0.935	0.982
Chittora et al. [11]	LSVM and SMOTE	0.988	1.000	-	-
Mienye and Sun [68]	Cost-sensitive RF	0.986	1.000	-	-
Akter et al. [69]	RFE and ANN	0.970	0.980	-	0.980
Akter et al. [69]	CFS and ANN	0.960	0.970	-	0.970
Qin et al. [9]	OSR and RF	0.995	0.993	-	-
Nishanth and Thiruvaran [13]	Enhanced LDA	0.980	0.960	-	-
Elkholy et al. [70]	DBN with Softmax classifier	0.985	0.875	-	-
Rashed-Al-Mahfuz et al. [71]	RF and FS	0.990	0.979	0.996	0.987
Silveira et al. [12]	Decision tree and SMOTE	0.989	0.990	-	-
Motwani et al. [14]	LR and RFE	0.983	0.990	-	-
Ogunleye and Wang [15]	XGBoost and ETC-US-RFE	0.976	1.000	0.917	0.990
This paper	Proposed CS AdaBoost	0.998	1.000	0.998	0.990

## Data Availability

Not applicable.

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
