# Peer review of "A Machine Learning Method with Filter-Based Feature Selection for Improved Prediction of Chronic Kidney Disease"

_bioengineering, 2022, doi:10.3390/bioengineering9080350_

Round 1

Reviewer 1 Report

 1. I would suggest summarizing in Table 1 biochemical data associated with 400 patient records; 250 correspond to patients with CKD, and 150 correspond to patients without CKD.

 2. I would also recommend add cut-off data that where used to distinguish between  ‘normal’ and ‘abnormal’ parameters.

3  3. In my opinion in CKD prediction studies such parameters as the duration of the disease and the age of onset may be of great importance and should be incorporated into statistical analyses.

Author Response

  1. We thank the reviewer for the suggested comments. We have modified the manuscript with more discussions about the biochemical data. Due to space constraints, we could not include it in Table 1, but it can be found in Lines 154-167 and 171-181.
  2. We thank the reviewer for the comment. The 'normal' and 'abnormal' categorization was done by the clinicians who prepared the data after examining the patients; hence, we do not have the cut-off data used. However, we have clarified the data categorization in the manuscript (Lines 198-206).
  3. We thank the reviewer for the comments and understand the importance of disease duration and the age of onset in CKD detection. However, the dataset does not contain those features. However, we have included it in Section 4 (Line 479-481), and we intend to incorporate both attributes in our future research.

Reviewer 2 Report

This is a well-designed, well-conducted study.  It designed a classifier with IG-enhanced feature selection and a cost-efficient boosting algorithm to classify chronic kidney diseases.  The classifier performance on imbalanced dataset was compared with and without the feature selection and in comparison to other popular classifiers.   High accuracy was achieved with new classifier.  The text was also well-written and easy to follow.

It is noted that the dataset was not CKD images, but binary values, and thus the applicability are limited to binary information.

Minor: Line 43, add (ML) after machine learning.

Author Response

We thank the reviewer for the comment. In the revised manuscript, we have made the recommended correction (Line 43).

Reviewer 3 Report

  A Machine Learning Method with Filter-based Feature Selection for Improved Prediction of Chronic Kidney Disease  

The proposed approach has produced an effective predictive model for CKD diagnosis and could be applied to more imbalanced medical datasets for effective
disease detection. What is the novelty of the proposed model? How is the proposed method effectively more efficient than others?  Introduction contains sufficient information but related work description is missing. Literature survey work needs to be strongly emphasized and also highlight the research issues and motivation of the proposed work.  How were the result values obtained?  Refer the following suitable and latest references as follows: Muthumanjula, M., and Ramasubramanian Bhoopalan. "Detection of White Blood Cell Cancer using Deep Learning using Cmyk-Moment Localisation for Information Retrieval." Journal of IoT in Social, Mobile, Analytics, and Cloud 4, no. 1 (2022): 54-72. Shakya, Subarna. "Modified Gray Wolf Feature Selection and Machine Learning Classification for Wireless Sensor Network Intrusion Detection." IRO Journal on Sustainable Wireless Systems 3, no. 2 (2021): 118-127.

Author Response

We thank the reviewer for the comments and suggestions to improve the manuscript. The recommended suggestions have been included in the revised manuscript. Below are specific comments regarding the corrections made.

  1. The study's novelty is in two parts: clinical and algorithmic contributions. Firstly, regarding the clinical contribution, the research employed the information gain technique to identify the most informative features in the CKD dataset that would result in enhanced classifier performance. Secondly, the algorithmic contribution of this study is that we developed a cost-sensitive classifier that gives more attention to the minority class samples, thereby leading to improved performance in both classes. Meanwhile, we have highlighted the contributions in the revised manuscript (Line 132-135).
  2. The proposed method is more effective since it achieves better classification performance, as seen in Table 4 and Fig. 3. Meanwhile, in the revised manuscript, we have highlighted the enhanced performance obtained by the proposed method (Line 407-414).
  3. We thank the reviewer for the observation and have improved the revised manuscript accordingly. Specifically, the introduction section has been updated to capture the suggestions (Lines 70-75, 78-91, 104-106, 114-116, and 125-129).
  4. The scikit-learning ML library was employed in developing the models, and in section 3, we have stated that (Line 325-328). Also, we referred to the suggested articles and included them in the revised manuscript, i..e. Ref 38 and Ref 43.

Reviewer 4 Report

A sound research for predicting CKD with a machine learning method.  A few suggestions for improvements:

1. Literature are loosely laid out and not well structured. Pros and Cons of existing methods are not discussed in a logic way to support the current research. In particular, feature selection needs to be justified. If all attributes are easily available from patients, why have to reduce features?

2. Why is accuracy not reported in the abstract?

3. The experiment first compared results with other ML methods, many of which are dated (refs 52-56). Why not directly compare with the more recent methods? It later did compare with ref 10-13. What about ref 14, 15?

4. The improvement of performance is not very significant from Table 2 and Figure 2 (compared to the dated methods).

Author Response

We thank the reviewer for the comments and suggestions to enhance the manuscript. The suggestions have been incorporated into the revised manuscript.

  1. We thank the reviewer for the observation. Section 1 has been revamped to capture the recommendations, which has significantly improved the manuscripts. The revisions can be found in Lines 80-82, 83-91, 104-106, 114-116, and 125-129.
  2. We thank the reviewer for noticing this oversight. We have included the accuracy in the abstract (Line 21).
  3. The ML methods in Refs 52-56 are the conventional ML methods, and they were used for different applications and not for CKD detection. Hence, we compared our method with these well-known ML methods first. Meanwhile, we have included Refs 14 and 15 in Table 5.
  4. We thank the reviewer for the comment. The proposed cost-sensitive AdaBoost obtained an increased performance compared to the traditional AdaBoost; for example, in Table 2, the accuracy increased by 3.7%, whereas the sensitivity, specificity, and AUC increased by 3.4%, 2.5%, and 3%. Secondly, the proposed model also performed better than the other ML classifiers. Also, the performance was further enhanced after the feature selection step, which is one of the research goals.

Round 2

Reviewer 1 Report

Accept in present form